# Myofibrillar Protein Profile of the Breast Muscle in Turkeys as a Response to the Variable Ratio of Limiting Amino Acids in Feed

Paweł Konieczka [1,2,*], Wiesław Przybylski [3], Danuta Jaworska [3], Elżbieta Żelechowska [3], Piotr Sałek [3], Dominika Szkopek [2], Aleksandra Drażbo [1], Krzysztof Kozłowski [1] and Jan Jankowski [1]

1 Department of Poultry Science and Apiculture, University of Warmia and Mazury in Olsztyn, Oczapowskiego 5, 10-719 Olsztyn, Poland
2 The Kielanowski Institute of Animal Physiology and Nutrition, Polish Academy of Sciences, Instytucka 3, 05-110 Jablonna, Poland
3 Institute of Human Nutrition Sciences, Warsaw University of Life Sciences, Nowoursynowska 159c, 02-776 Warsaw, Poland
* Correspondence: pawel.konieczka@uwm.edu.pl

**Abstract:** The effects of the different dietary levels of Arginine (Arg) in low- and high-methionine (Met) diets on the meat quality and myofibrillar protein profile of breast muscles from turkeys were determined. The experiment had a completely randomized $3 \times 2$ factorial design with three levels of Arg (90%, 100%, and 110%) relative to the dietary Met levels (30% or 45%). At 42 days of age, eight turkeys from each treatment were sacrificed; the meat pH value was measured at 48 h post-slaughter, and meat color was measured according to the CIE L*a*b* system. The SDS-PAGE method was performed to investigate the myofibrillar protein profile of the breast muscle. The analysis of variance showed a significant effect of the Arg or Met dietary levels on the color parameter b* and the profile of myofibrillar proteins in muscles. The results of the cluster analysis of the myofibrillar protein profile showed that, with a high level of Arg (i.e., 110%), the level of Met 35% or 45% was less important. It can be concluded that the increase in the share of Arg and Met in the diet of turkeys increases the content of some myofibrillar proteins (actinin, desmin, actin) and reduces degradation during the post-slaughter proteolysis of proteins that are considered tenderization indicators.

**Keywords:** poultry; amino acids; arginine; methionine; myofibrillar protein

## 1. Introduction

In modern poultry nutrition, the quality of the protein used is very important in terms of its biological value, i.e., its amino acid composition and mainly the content of essential amino acids. Among the amino acids that limit the biological value of protein, arginine (Arg) and methionine (Met) play a very important role [1–7]. Poultry are susceptible to a dietary deficiency of Arg because they are incapable of its synthesis de novo due to lacking a functional urea biochemical cycle [1,3]. Arginine and methionine are precursors for other essential molecules in immune defense, the antioxidant system, cell signaling, and gene expression, and they can act as regulators in the growth and development of the animals [7–9]. Castro and Kim [8] emphasized that Arg supplementation has been shown to modulate musculoskeletal health development, can reduce fat accretion, and can improve the antioxidant system. Arginine also influences the secretagogue, stimulating the release of growth hormone, insulin, and insulin-like growth factor in the bloodstream, as well as body weight, carcass yield, and reduced fat deposition, and the formation of collagen and connective tissue [3,10]. Additionally, on the basis of research, it can be concluded that Arg could influence skeletal muscle synthesis and reduce protein degradation [1,3,9,11]. However, this interaction has not been thoroughly studied. On the other hand, Met is recognized as a precursor in the biosynthesis of various metabolites (sarcosine, betaine, and choline via transmethylation) and as an intermediary in the conversion to cystine or

cysteine (via homocysteine) [2,12]. Moreover, Met could improve bone development and potentially mitigate the negative effects caused by heat stress [13,14]. Understanding how these amino acids can ameliorate stressful conditions may provide novel insights into their use as nutritional strategies to modulate the health status of birds [8,15]. Ruth and Field [5] postulated that the use of Met in compound feed in excess of the recommended levels decreased oxidative stress and reduced bacterial infections. It should be pointed out that Lys, Met, and Arg are amino acids that limit the biological value of protein in cereal-soybean meal-based diets for turkeys [16], but the determination of their optimal dietary inclusion rates and ratios stirs much controversy [4,17,18]. The effect of different levels of Arg and Met on production results, such as growth rate, weight gain, and metabolic indicators and immune processes, has already been extensively studied [4,7]. However, there is no research on how the variable share of these amino acids in the feed ration may affect the profile of poultry muscle proteins and their post-slaughter proteolysis, which may affect the quality of meat and its functional properties related to processing and sensory quality. We additionally speculate that the dietary profile of essential amino acids may have a great contribution in this regard in the early stage of the turkey's growth. Therefore, in this study, it was hypothesized that the varying proportion and ratio of Arg and Met in the feed ration will affect the profile of myofibrillar proteins in turkey muscles and the degree of their proteolysis during meat maturation after slaughter.

## 2. Materials and Methods

### 2.1. Birds and Housing

One-day-old Hybrid Converter female turkey poults (216 in total), obtained on the day of hatching from a commercial hatchery (Grelavi Company in Ketrzyn, NE, Poland), were randomly allocated to 12 pens with litter (4 m$^2$ each; 2.0 m × 2.0 m) considering their average initial body weight, and they were reared over the 42-day experimental period. The birds were reared under the following conditions: temperature (32 °C for the 1st week, 29 °C for the 2nd week, 28 °C for the 3rd week, 26 °C for the 4th week, 23 °C for the 5th week, and 21 °C for the 6th week), humidity (maintained less than 60%), and lighting program (24 h of light over the first 2 days, then 1 h of light from day 3 to day 14, then 2 h of light between day 15 and day 21, then 4 h of light between day 22 and day 35, and then 6 h of light between day 36 and day 42). The maintained conditions were the same for all pens located in the experimental room and followed the recommendations for standard management practices for Hybrid Converter turkeys [19].

### 2.2. Diets and Experimental Design

The total content of Lys, Arg, and Met of experimental diets was previously reported by Jankowski et al. [4] (Table 1), whereas detailed feeding programs, as well as the composition of the experimental diets, were previously reported by Konieczka et al. [7] (Table 2). In brief, birds received experimental diets ad libitum, which were formulated to be isocaloric and to meet or exceed the turkeys' nutritional needs. Diets with high levels of Lys contained approximately 1.80% and 1.65% Lys, respectively, in two successive feeding periods (days 1–28 and days 29–42). The supplementary levels of Lys were consistent with the nutritional specifications established in the Management Guidelines for Raising Commercial Turkeys at respective ages [19]. The experiment had a completely randomized 3 × 2 factorial design with three levels of Arg (90%, 100%, and 110%) relative to the content of dietary Met (30% or 45%). The designed level of specific amino acids in the diets was achieved by the analytical determination of the basal diet regarding its concentration and supplementation appropriate amount thereafter, and the final concentration was again confirmed analytically.

**Table 1.** Total content of lysine (Lys), methionine (Met), and arginine (Arg) of experimental diets fed to turkeys in two successive periods (g/100 g) [1].

| Treatment [2] | Feeding Periods, Days | | | | | |
| | From 1 to 28 | | | From 29 to 42 | | |
| | Lys | Arg | Met | Lys | Arg | Met |
|---|---|---|---|---|---|---|
| $Arg_{90}Met_{30}$ | 1.63 | 1.46 | 0.50 | 1.48 | 1.37 | 0.42 |
| $Arg_{90}Met_{45}$ | 1.56 | 1.43 | 0.69 | 1.45 | 1.39 | 0.66 |
| $Arg_{100}Met_{30}$ | 1.58 | 1.52 | 0.51 | 1.53 | 1.53 | 0.42 |
| $Arg_{100}Met_{45}$ | 1.66 | 1.56 | 0.71 | 1.56 | 1.56 | 0.7 |
| $Arg_{110}Met_{30}$ | 1.55 | 1.69 | 0.52 | 1.56 | 1.71 | 0.44 |
| $Arg_{110}Met_{45}$ | 1.64 | 1.67 | 0.74 | 1.55 | 1.73 | 0.66 |

[1] Adopted from Jankowski et al. [4]. [2] Dietary treatments contained three levels of arginine (90%; Arg90, 100%; Arg100, and 110%; Arg110) relative to the content of dietary methionine (30%; Met30 or 45%; Met45).

**Table 2.** Composition and nutrient content of basal diets (g/100 g) [1].

| Ingredients | Feeding Periods, Days | |
| | From 1 to 28 | From 29 to 42 |
|---|---|---|
| Wheat | 43.98 | 47.72 |
| Maize | 10.00 | 10.00 |
| Soybean meal | 28.77 | 26.54 |
| Rapeseed meal | 3.00 | 3.00 |
| Potato protein | 5.00 | 5.00 |
| Soibean oil | 0.95 | 2.85 |
| Maize gluten meal | 3.50 | 3.00 |
| Sodium bicarbonate | 0.20 | 0.20 |
| Sodium chloride | 0.15 | 0.16 |
| Limestone | 2.07 | 1.87 |
| Monocalcium phosphate | 1.94 | 1.55 |
| L-Threonine | 0.09 | 0.10 |
| Choline chloride | 0.10 | 0.10 |
| Vitamin-mineral premix [2] | 0.25 | 0.25 |
| Calculated nutrients content | | |
| Metabolizable energy, kcal/kg | 2820 | 2950 |
| Crude potein | 27.0 | 24.5 |
| Arginine | 1.58 | 1.44 |
| Lysine | 1.36 | 1.19 |
| Methionine | 0.44 | 0.39 |
| Met + Cys | 0.91 | 0.83 |
| Threonine | 1.02 | 1.01 |
| Calcium | 1.30 | 1.15 |
| Avaiable phosphorus | 0.70 | 0.60 |

[1] Adopted from Konieczka et al. [7]. [2] Provided per kg diet (feeding periods: weeks 0−4 and 5−6): mg: retinol 3.78 and 3.38, cholecalciferol 0.13 and 0.12, a-tocopheryl acetate 100 and 90, vit. K3 5.8 and 5.6, thiamine 5.4 and 4.7, riboflavin 8.4 and 7.5, pyridoxine 6.4 and 5.6, cobalamin 0.032 and 0.028, biotin 0.32 and 0.28, pantothenic acid 28 and 24, nicotinic acid 84 and 75, folic acid 3.2 and 2.8, Fe 64 and 60, Mn 120 and 112, Zn 110 and 103, Cu 23 and 19, I 3.2 and 2.8, Se 0.30 and 0.28, respectively.

*2.3. Sample Collection*

At 42 days of age, eight turkeys representing the group's average body weight from each treatment were sacrificed by cervical dislocation, and the right breast muscle (pectoralis major) was collected, placed in a plastic bag separately, and transported to the laboratory in polystyrene bags in cold conditions for analyses. The samples were then chilled for 24 h at 4 °C. The next day, meat quality parameters were evaluated in the samples at the laboratory. The samples for the extraction of muscle protein were packed in closed Ziploc bags and frozen at −80 °C. The samples were stored for up to one month for further analysis.

### 2.4. Meat Quality Measurement

The meat pH value was measured at 48 h (pH 48) after slaughter with a WTW 330i pH meter (Weilheim, Germany) equipped with special electrodes (SenTix®SP Number 103645) to measure pH directly in meat (three measurements). The equipment was calibrated using WTW buffers with pH values of 4.01 (No. 108706) and 7.00 (No. 108708), consistent with NIST/PTB (National Institute of Standards and Technology/Polish Biochemical Society).

Meat color was measured in the CIE L*a*b* system using a CR310 Minolta Chroma Meter (Osaka, Japan) with a D65 light source at 8° standard observer and 8 mm aperture at 48 h post-mortem. The meat samples (length of 2 cm) were cut and bloomed for 1 h at 4 °C with no surface covering prior to color measurements (in triplicate).

### 2.5. Extraction of Muscle Protein

A meat sample of 150 mg (from the middle part of the muscle) was homogenized at 13,500 rpm with 600 µL of 0.003 M Phosphate Buffer at pH 7 for 1 min. The homogenate was centrifuged at $10,000 \times g$ for 3 min. The supernatant of sarcoplasmic proteins was aliquoted and frozen at −80 °C. The pellet of myofibrillar protein was resuspended with 1 mL of denaturing solution containing: 8.3 M Urea, 2 M Thiourea, 64 mM Dithiothreitol (DTT), cholamidopropyldimethyl hydroxypropane sulfonate (CHAPS) 2%, NP-40 2%, Glycerol 10%- and 20-mM Tris-HCl, pH 8. The samples were kept in contact overnight. The next day, they were homogenized at 9500 rpm for 90 s. After homogenization, the supernatant was aliquoted and stored at −80 °C for further analysis. Protein concentrations were determined by Bradford protein assay (Roti®-Quant) using bovine serum albumin (BSA) as a standard.

### 2.6. Sodium Dodecyl Sulphate Polyacrylamide Gel Electrophoresis (SDS-PAGE)

The SDS-PAGE of myofibrillar protein was performed according to the method of Bollag and Edelstein [20] using the BioRad apparatus. Proteins were resolved on a 12% separation gel and 5% stacking gel. Gels were first run for approximately 1 h at 75 V, followed by 6 h at 150 V. Gels were stained with Coomassie Brilliant Blue R250. Protein markers (Thermo Scientific[TM] (Waltham, CA, USA) Page Ruler[TM] Unstained Protein Ladder) were used for molecular weight determination. Image analysis and quantification were performed using GelScan v. 1.45 software (Kucharczyk TE, Warsaw, Poland).

### 2.7. Statistical Analyses

The obtained data were developed using Statistica version 13 software [21]. The basic descriptive statistics (mean, standard error) were calculated. The normality of data distribution was checked by the Shapiro-Wilk test. The effects of the influence of three levels of Arg (90%, 100%, and 110%) relative to the two levels of Met (30% or 45%) were evaluated using two-way ANOVA ($3 \times 2$ factorial design). The significance of differences between means was determined based on the least significant differences test (LSD). All significances were tested at the level of $p < 0.05$. A principal component analysis (PCA) and Cluster analysis (CA) as a multivariate analysis was performed to gain a better understanding of the total variability and characteristics of all experimental groups with respect to the myofibrillar protein profile.

## 3. Results

*Effects of Dietary Treatments on Bird's Performance*

The final body weight (BW), body weight gain (BWG), daily feed intake (DFI), and feed conversion ratio (FCR) recorded for the experimental period of 1–42 days are presented in Table 3. The body weight of turkeys at 42 days of life averaged $2.42 \pm 0.061$ kg, and it was not significantly affected due to the dietary Met levels ($p = 0.521$). However, the highest dietary Arg level resulted in significantly higher body weight in comparison to Arg90 (2.50 vs. 2.33 kg, $p = 0.004$). A significant interaction ($p = 0.014$) was recorded between Arg90 and Arg110, as the Arg110 increased body weight in combination with Met45, whereas the

opposite was true for Arg90. The remaining investigated performance indices, including BWG, DFI, and FCR, did not differ between treatments, and no significant interactions (Arg levels × Met levels) were found ($p > 0.05$).

**Table 3.** Final body weight (BW), body weight gain (BWG), average daily feed intake (DFI), and feed conversion ratio (FCR) in turkeys at 6 weeks of age.

| Treatment | BW, kg | BWG, kg | DFI, g/day | FCR, kg/kg |
|---|---|---|---|---|
| Arginine (Arg) level, % (A) | | | | |
| 90 | 2.33 [a] | 2.47 | 91.4 | 1.64 |
| 100 | 2.43 [ab] | 2.46 | 91.7 | 1.65 |
| 110 | 2.50 [b] | 2.46 | 91.8 | 1.65 |
| Methionine (Met) level, %, (B) | | | | |
| 30 | 2.41 | 2.45 | 90.8 | 1.64 |
| 45 | 2.44 | 2.47 | 92.4 | 1.66 |
| SEM | 0.023 | 0.016 | 0.732 | 0.006 |
| *p*-value | | | | |
| Arg | 0.004 | 0.993 | 0.967 | 0.713 |
| Met | 0.521 | 0.518 | 0.305 | 0.142 |
| Interactions | | | | |
| A × B | 0.014 | 0.934 | 0.900 | 0.694 |

[a,b] values in same column with no common superscript show a significant difference ($p \leq 0.05$).

The data for the meat quality are summarized by dietary treatment in Tables 4 and 5.

**Table 4.** Breast meat quality characteristics in relation to the main effect of different arginine (Arg) to methionine (Met) ratios in the diet of turkeys at 42 days of age.

| Traits | Dietary Treatments [1] | | | | | | | |
|---|---|---|---|---|---|---|---|---|
| | Arg (%) Level | | | | Met (%) Level | | | |
| | **90** | **100** | **110** | *p*-Value | **30** | **45** | *p*-Value | **SEM** |
| pH | 5.68 | 5.68 | 5.67 | 0.95 | 5.68 | 5.67 | 0.69 | 0.01 |
| Color L* | 49.62 | 50.85 | 50.47 | 0.27 | 50.33 | 50.3 | 0.96 | 0.31 |
| Color a* | −0.51 | −0.32 | −0.10 | 0.52 | −0.23 | −0.39 | 0.57 | 0.15 |
| Color b* | 5.84 | 6.24 | 6.45 | 0.68 | 6.70 | 5.65 | 0.07 | 0.32 |

[1] Dietary treatments contained three levels of arginine (90%; Arg90, 100%; Arg100, and 110%; Arg110) relative to the content of dietary methionine (30%; Met30 or 45%; Met45); L*, a*, b* - Meat color was measured in to the CIE L*a*b* system.

**Table 5.** Breast meat quality characteristics in relation to the interaction effect of different arginine (Arg) to methionine (Met) ratios in the diet of turkeys at 42 days of age.

| Traits | Nutritional Effect of Arg–Met Ratio | | | | | | | SEM |
|---|---|---|---|---|---|---|---|---|
| | Met 30(%) Level | | | Met 45(%) Level | | | | |
| | Arg (%) Level | | | Arg (%) Level | | | *p*-Value | |
| | **90** | **100** | **110** | **90** | **100** | **110** | | |
| pH | 5.70 | 5.68 | 5.67 | 5.66 | 5.68 | 5.68 | 0.71 | 0.01 |
| Color L* | 48.93 | 50.97 | 51.09 | 50.31 | 50.74 | 49.85 | 0.24 | 0.31 |
| Color a* | −0.88 | 0.03 | 0.16 | −0.15 | −0.66 | −0.37 | 0.11 | 0.15 |
| Color b* | 5.04 [a] | 7.39 [b] | 7.68 [b] | 6.65 [ab] | 5.09 [a] | 5.20 [a] | 0.05 | 0.32 |

[a,b] values in same column with no common superscript show a significant difference ($p \leq 0.05$); L*, a*, b*—Meat color was measured in to the CIE L*a*b* system.

The analysis of variance did not show a significant effect of the Arg or Met content on the pH or color parameters, except for a significant interaction effect for the color parameter b* (Tables 4 and 5).

The results presented in Table 5 showed an interaction effect, and it showed that, as the level of Arg increased at a level of Met of 30%, the color tone of the meat appeared more yellow. On the other hand, the opposite situation was observed when the Met level was 45%, in which case the meat color shade was increasingly less yellow (Table 5).

The main effects of different Arg and Met levels in turkey diets on the myofibrillar protein profile are presented in Figure 1 and in Tables 6 and 7. The analysis of the main effects presented in Table 6 shows that the level of Arg in the ration significantly affected the level of the following proteins: myosin HC, 140 kDa; α-actinin, 60 kDa; desmin, 45 kDa; tropomyosin, 34 kDa, 33 kDa, 32 kDa, 28 kDa, 26 kDa, and 14 kDa. The results of the research showed that, with the increase in the level of Arg in the diet from 90% to 110%, a significant increase in the level of such proteins as 140 kDa, α-actinin, desmin, 45 kDa, and 34 kDa was observed. On the other hand, other proteins such as myosin HC, 60 kDa; tropomyosin, 33 kDa, 32 kDa, 28 kDa, 26 kDa, and 14 kDa have been shown to decrease with increasing Arg in the diet. In turn, the influence of the main effect of the level of Met in the diet was found for the levels of myosin HC, desmin, tropomyosin, TnC, and myosin LC2 (Table 6). For the first three proteins (myosin HC, desmin, and tropomyosin), it was observed that an increase in the level of Met in the diet from 30% to 45% resulted in a decrease in the content of these proteins, while the opposite situation was observed for TnC and myosin LC2, whose levels increased as a result of increasing the content of Met in the ratio (Table 6). The simultaneous impact of different levels in the diet of Arg and Met on the profile of myofibrillar proteins, as the result of their interaction study, is presented in Table 7.

**Table 6.** Characteristics of the myofibrillar protein profile of breast meat in relation to the main effect of different arginine (Arg) to methionine (Met) ratios in - age (main effect).

| Traits | Dietary Treatments [1] | | | | | | | |
| | Arg (%) Level | | | | Met (%) Level | | | |
| | **90** | **100** | **110** | *p*-Value | **30** | **45** | *p*-Value | **SEM** |
|---|---|---|---|---|---|---|---|---|
| 1-Miosin HC | 11.40 [a] | 10.87 [ab] | 10.27 [b] | 0.05 | 11.19 [a] | 10.49 [b] | 0.03 | 0.19 |
| 2–140 kDa | 7.46 [a] | 7.72 [a] | 8.52 [b] | 0.05 | 7.79 | 8 | 0.30 | 0.13 |
| 3-α-actinin | 3.90 [a] | 4.03 [a] | 4.49 [b] | 0.05 | 3.99 | 4.28 | 0.07 | 0.09 |
| 4-α-actinin | 4.5 | 4.53 | 4.32 | 0.42 | 4.44 | 4.46 | 0.91 | 0.08 |
| 5–60 kDa | 5.15 [a] | 4.81 [a] | 4.33 [b] | 0.05 | 4.92 | 4.61 | 0.09 | 0.11 |
| 6–55 kDa | 0.79 | 0.96 | 0.98 | 0.41 | 0.84 | 0.98 | 0.25 | 0.06 |
| 7-Desmin | 1.50 [a] | 1.66 [a] | 1.95 [b] | 0.05 | 1.83 [a] | 1.58 [b] | 0.05 | 0.06 |
| 8–48 kDa | 3.83 | 3.82 | 3.95 | 0.73 | 3.95 | 3.78 | 0.28 | 0.07 |
| 9–45 kDa | 0.66 [a] | 1.11 [a] | 1.83 [b] | 0.05 | 1.16 | 1.24 | 0.76 | 0.16 |
| 10-Actin | 17.3 | 17.65 | 18.38 | 0.09 | 17.96 | 17.59 | 0.36 | 0.22 |
| 11-TnT | 5.15 | 4.99 | 5.55 | 0.09 | 5.26 | 5.2 | 0.77 | 0.11 |
| 12-Tropomyosin | 7.41 [a] | 7.68 [a] | 6.14 [b] | 0.05 | 7.45 [a] | 6.70 [b] | 0.05 | 0.18 |
| 13–34 kDa | 0.54 [a] | 0.67 [a] | 2.59 [b] | 0.05 | 1.1 | 1.43 | 0.90 | 0.10 |
| 14–33 kDa | 4.18 [a] | 4.29 [a] | 3.71 [b] | 0.05 | 4.06 | 4.06 | 0.96 | 0.07 |
| 15–32 kDa | 2.48 [a] | 2.36 [a] | 1.56 [b] | 0.05 | 2.18 | 2.09 | 0.52 | 0.10 |
| 16–30 kDa | 0.83 | 1.03 | 1.06 | 0.35 | 0.96 | 0.99 | 0.88 | 0.07 |
| 17–28 kDa | 3.15 [a] | 2.60 [b] | 2.28 [b] | 0.05 | 2.58 | 2.77 | 0.32 | 0.10 |
| 18–26 kDa | 1.06 [a] | 1.03 [a] | 0.74 [b] | 0.05 | 0.93 | 0.96 | 0.62 | 0.06 |
| 19-Myosin LC1 | 4.06 | 4.01 | 3.72 | 0.07 | 3.81 | 4.05 | 0.07 | 0.08 |
| 20-TnI | 4.1 | 4.01 | 3.88 | 0.58 | 3.84 | 4.16 | 0.08 | 0.10 |
| 21-TnC | 1.84 | 1.53 | 1.64 | 0.15 | 1.51 [a] | 1.83 [b] | 0.05 | 0.07 |

**Table 6.** *Cont.*

| Traits | Dietary Treatments [1] | | | | | | | |
|---|---|---|---|---|---|---|---|---|
| | Arg (%) Level | | | | Met (%) Level | | | |
| | **90** | **100** | **110** | ***p*-Value** | **30** | **45** | ***p*-Value** | **SEM** |
| 22-Myosin LC2 | 3.77 | 4.41 | 3.87 | 0.19 | 3.67 [a] | 4.36 [b] | 0.05 | 0.19 |
| 23–16 kDa | 2.67 | 2.16 | 2.39 | 0.10 | 2.43 | 2.39 | 0.80 | 0.10 |
| 24–14 kDa | 2.23 [a] | 2.05 [ab] | 1.83 [b] | 0.05 | 2.09 | 1.99 | 0.37 | 0.06 |

[1] Dietary treatments contained three levels of arginine (90%; Arg90, 100%; Arg100, and 110%; Arg110) relative to the content of dietary methionine (30%; Met30 or 45%; Met45). [a,b]—Mean values followed by different capital superscript letters are significantly different at the $p \leq 0.05$ level.

**Table 7.** Characteristics of the myofibrillar protein profile of breast meat in relation to the interaction effect of different arginine (Arg) to methionine (Met) ratios in the diet of turkeys at 42 days of age.

| Traits | Nutritional Effect of Arg × Met [1] | | | | | | ***p*-Value** | **SEM** |
|---|---|---|---|---|---|---|---|---|
| | Met 30 (%) Level | | | Met 45 (%) Level | | | | |
| | Arg (%) Level | | | Arg (%) Level | | | | |
| | **90** | **100** | **110** | **90** | **100** | **110** | | |
| 1-Miosin HC | 11.15 [ab] | 11.78 [a] | 10.65 [bc] | 11.65 [ab] | 9.95 [c] | 9.88 [c] | 0.05 | 0.19 |
| 2-140 kDa | 7.62 [ac] | 7.21 [a] | 8.55 [b] | 7.29 [a] | 8.21 [bc] | 8.49 [b] | 0.05 | 0.13 |
| 3-α-actinin | 3.90 | 3.61 | 4.47 | 3.90 | 4.45 | 4.50 | 0.06 | 0.09 |
| 4-α-actinin | 4.86 [a] | 4.46 [abc] | 4.00 [b] | 4.13 [bc] | 4.60 [ac] | 4.65 [a] | 0.05 | 0.08 |
| 5–60 kDa | 5.44 | 5.08 | 4.25 | 4.85 | 4.55 | 4.42 | 0.17 | 0.11 |
| 6–55 kDa | 0.93 | 0.80 | 0.79 | 0.66 | 1.11 | 1.16 | 0.07 | 0.06 |
| 7-Desmin | 1.60 | 1.91 | 2.00 | 1.42 | 1.42 | 1.90 | 0.32 | 0.06 |
| 8–48 kDa | 3.75 | 4.09 | 3.99 | 3.90 | 3.54 | 3.90 | 0.17 | 0.07 |
| 9–45 kDa | 0.68 [a] | 0.56 [a] | 2.25 [b] | 0.65 [a] | 1.66 [b] | 1.41 [ab] | 0.05 | 0.16 |
| 10-Actin | 18.08 [ab] | 17.08 [ac] | 18.73 [b] | 16.53 [c] | 18.22 [ab] | 18.04 [ab] | 0.05 | 0.22 |
| 11-TnT | 5.15 | 4.76 | 5.87 | 5.15 | 5.21 | 5.24 | 0.13 | 0.11 |
| 12-Tropomyosin | 7.38 [ab] | 7.92 [b] | 7.06 [a] | 7.44 [ab] | 7.45 [ab] | 5.22 [c] | 0.05 | 0.18 |
| 13–34 kDa | 0.53 | 0.68 | 2.11 | 0.55 | 0.67 | 3.06 | 0.21 | 0.10 |
| 14–33 kDa | 4.29 | 4.15 | 3.74 | 4.07 | 4.43 | 3.67 | 0.28 | 0.07 |
| 15–32 kDa | 2.38 [ad] | 2.69 [a] | 1.47 [b] | 2.59 [a] | 2.02 [cd] | 1.65 [bc] | 0.05 | 0.1 |
| 16–30 kDa | 0.79 | 1.16 | 0.94 | 0.86 | 0.90 | 1.18 | 0.36 | 0.07 |
| 17–28 kDa | 2.72 [ab] | 2.85 [b] | 2.17 [a] | 3.58 [c] | 2.35 [ab] | 2.38 [ab] | 0.05 | 0.10 |
| 18–26 kDa | 0.88 [a] | 1.32 [c] | 0.57 [b] | 1.25 [c] | 0.74 [ab] | 0.90 [a] | 0.05 | 0.06 |
| 19-Myosin LC1 | 3.66 [ab] | 4.31 [cd] | 3.46 [a] | 4.46 [d] | 3.70 [ab] | 3.98 [bc] | 0.05 | 0.08 |
| 20-TnI | 3.82 [ab] | 4.23 [a] | 3.47 [b] | 4.39 [a] | 3.79 [ab] | 4.29 [a] | 0.05 | 0.10 |
| 21-TnC | 1.65 | 1.46 | 1.42 | 2.02 | 1.61 | 1.86 | 0.62 | 0.07 |
| 22-Myosin LC2 | 3.98 [a] | 3.28 [a] | 3.77 [a] | 3.56 [a] | 5.54 [b] | 3.98 [a] | 0.05 | 0.19 |
| 23–16 kDa | 2.34 [ab] | 2.54 [a] | 2.42 [ab] | 3.00 [a] | 1.79 [b] | 2.37 [ab] | 0.05 | 0.10 |
| 24–14 kDa | 2.39 | 2.06 | 1.83 | 2.07 | 2.05 | 1.84 | 0.41 | 0.06 |

[1] Dietary treatments contained three levels of arginine (90%; Arg90, 100%; Arg100, and 110%; Arg110) relative to the content of dietary methionine (30%; Met30 or 45%; Met45). [a,b,c,d] Mean values followed by different capital superscript letters are significantly different at the $p \leq 0.05$ level.

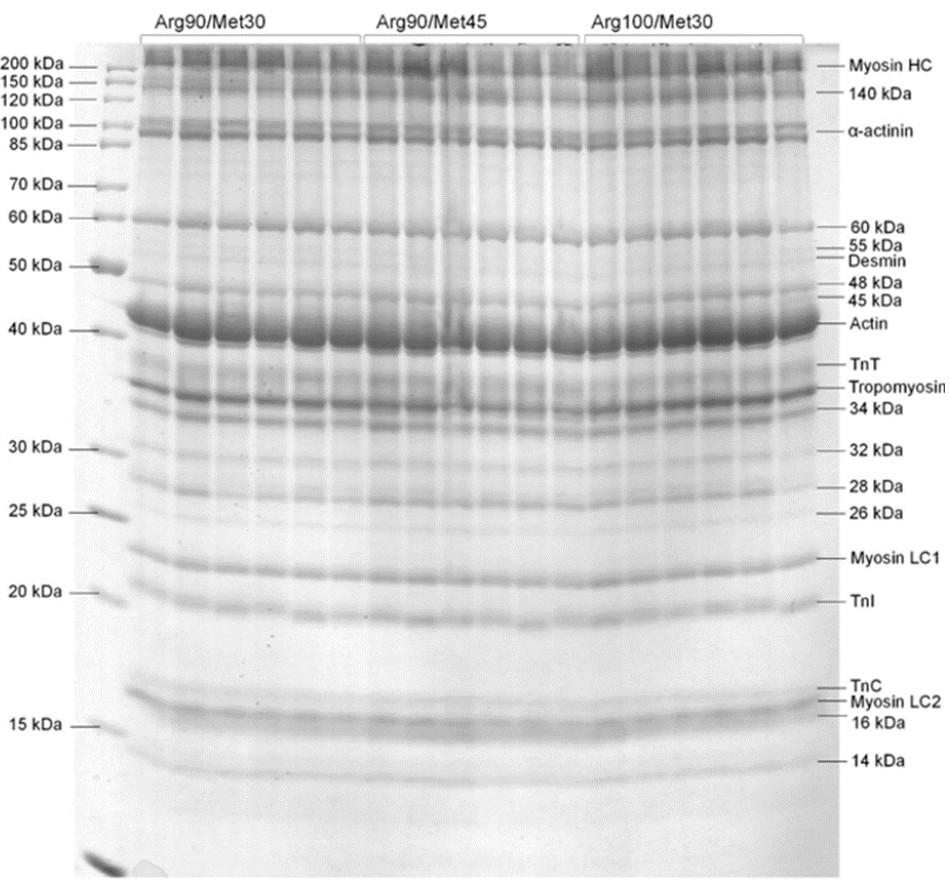

**Figure 1.** SDS-PAGE of muscle protein isolated from the pectoralis muscle. The left lane corresponds to the molecular weight scale. The following abbreviations are used: TnT troponin T, TnI troponin I, and TnC troponin C. Birds were fed dietary treatments that contained three levels of arginine (90%; Arg90, 100%; Arg100, and 110%; Arg110) relative to the content of dietary methionine (30%; Met30 or 45%; Met45).

A decrease or increase in the level, similar to the main effects, was found for such proteins as myosin HC, 140 kDa, α-actinin, 45 kDa; tropomyosin, 32 kDa, 28 kDa, and 26 kDa (Table 7). On the other hand, completely new effects in the study of interactions were found for such proteins as actin, myosin LC1, TnI, and 16 kDa (Table 7). In the case of actin, at a low Met level of 30%, only a high addition of Arg (110%) caused a significant increase in the amount of this protein. However, at a higher Met level (45%), a significantly higher amount of actin was found from the Arg level of 100 and 110% (Table 7). A completely different and difficult-to-understand relationship was found in the case of the levels of myosin LC1, LC2, and TnI. These data indicate a different type of proteolysis of these proteins depending on the level of Arg and Met in the birds' diet.

Multivariate statistical analyses, i.e., principal component analysis and cluster analysis, showed that individual groups had their own unique and characteristic protein profile (Figures 1–3). However, among the studied groups, the groups in which the level of Arg was 110%, regardless of the level of Met in the diet, definitely stood out. These results, therefore, indicate that the level of Arg and Met in the diet significantly affects the profile of myofibrillar proteins in muscles, which may be the result of differential proteolysis, as well as a different protein profile before slaughter.

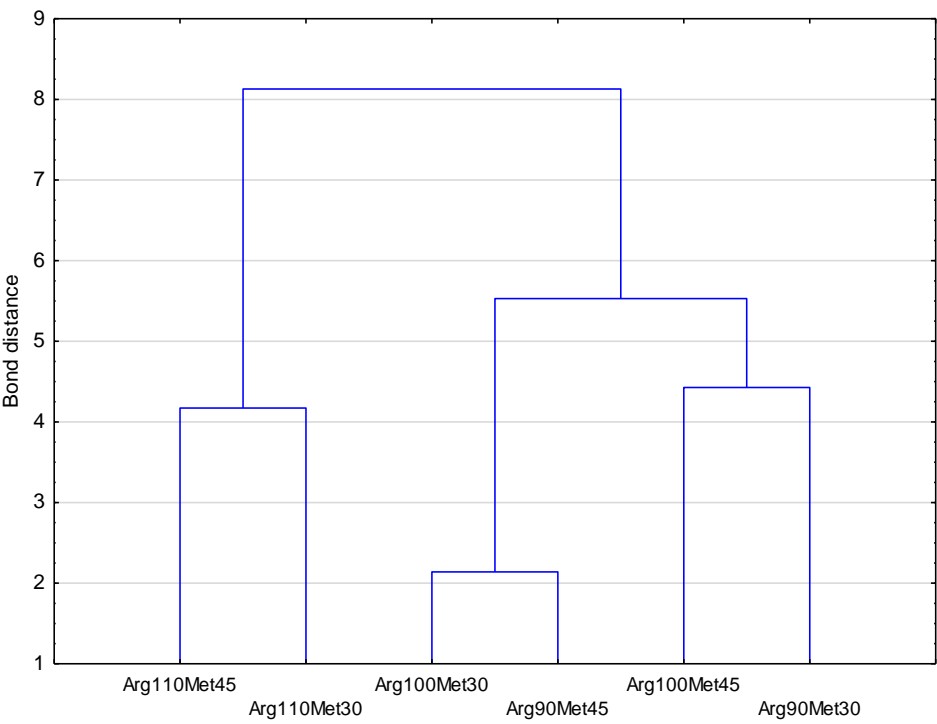

**Figure 2.** Results of the cluster analysis of the myofibrillar protein profile of turkey breast meat by groups. Birds were fed dietary treatments that contained three levels of arginine (90%; Arg90, 100%; Arg100, and 110%; Arg110) relative to the content of dietary methionine (30%; Met30 or 45%; Met45).

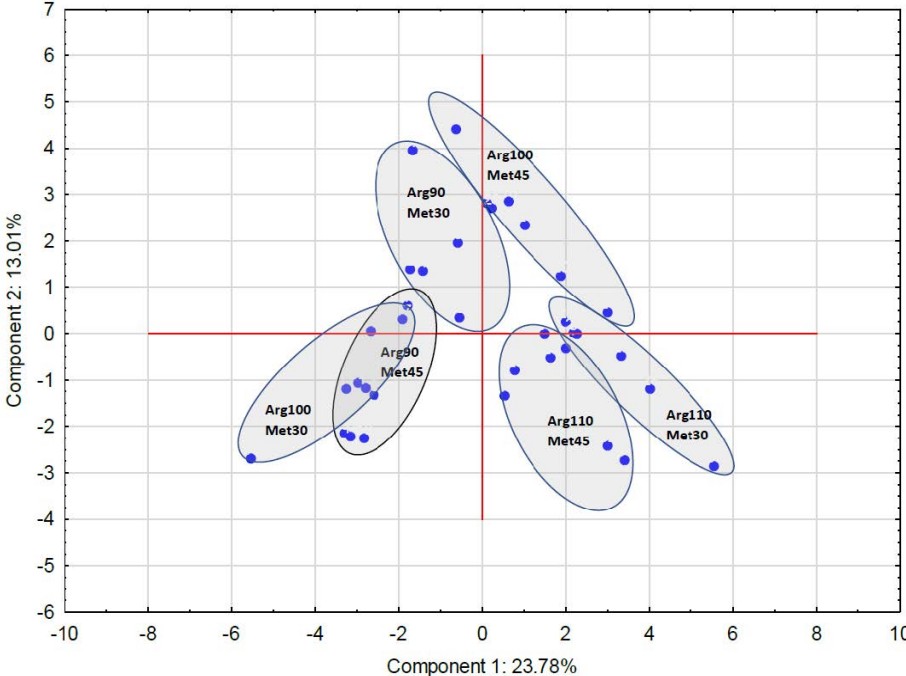

**Figure 3.** Location of the studied groups in clusters and in the space created by the first two components as the result of the principal component analysis. The arrangement of the groups reflects the grouping shown in Figure 2.

Birds were fed dietary treatments that contained three levels of arginine (90%; Arg90, 100%; Arg100, and 110%; Arg110) relative to the content of dietary methionine (30%; Met30 or 45%; Met45).

## 4. Discussion

The performance response of turkeys as a result of treatments indicated that different levels of Arg but not Met significantly affected the birds' performance. Overall, the final body weight at day 42 was 5.1% lower than that expected for commercial Hybrid Converter female turkeys at 42 days of age [19].

The average values of pH and color parameters presented in this study were lower then presented by Çelen et al. [22] and were similar to the range of pH presented by Rammouz et al. [23]. The results of the research conducted by Wynven et al. [24] showed that lower final pH values in turkey meat were associated with increased drip loss from muscle tissue and changes in color parameters in the L*a*b* system. As shown by Rammouz et al. [23], a lower pH value was associated with a higher glycolytic potential of muscles and, according to Çelen et al. [22], with the appearance of PSE meat. At the same time, it should be mentioned that Zampiga et al. [25] showed a lower final pH in the meat of turkeys fed a diet with a higher Arg content compared to the commercially used diet.

Our study shows that an increase in the proportion of arginine in relation to different Met levels could affect the yellowness of the meat. Some studies have shown that changes in the yellowness (b* parameter) appear in the case of defects in poultry meat. Pekel et al. [26] showed lower yellowness in the case of white striping in broiler chickens. In another study, Bowker and Zhuang [27] showed lower yellowness in broiler breast filets with higher water-holding capacity. The relationship between yellowness and drip loss of breast muscle was also confirmed by Rammouz et al. [23]. Cai et al. [28] and Zhang et al. [29] showed higher b* values in meat with wooden breasts of broilers in comparison to normal meat. Tasoniero et al. [30] also showed higher b* values in spaghetti meat in comparison to normal meat. Baldi et al. [31] observed that an increase in the yellowness of muscle is related to adipose tissue accumulation among muscle fibers occurring in spaghetti meat. Dalle Zotte et al. [32] stated that fat possesses a yellowish color and might be directly contributed to the increased yellowness of meat. Moreover, Hocquette et al. [33] stated that an increase in yellow color is associated with fat deposition in meat. On the other hand, Lee and Choi [34] found no difference in the yellowness of PSE and white-stripping muscles compared to normal-quality chicken meat. According to Jankowski et al. [4], arginine as a precursor in the synthesis of various metabolites could influence muscle metabolism and then, in consequence, could influence the meat color b* parameter. The obtained results in relation to pH and other parameters of color are in agreement with the results of Zampiga et al. [35] and Jankowski et al. [4], who also did not note the effect of increasing arginine levels in feed on meat quality parameters, such as pH or color a* parameter in chickens and turkeys.

These results, according to different profiles of myofibrillar protein in studied groups, confirm previous studies showing that the level of Arg and Met in the diet affect the sarcoplasmic protein profile (especially glycolytic enzymes) in breast muscle and thus may have significant effects on muscle cells [7]. However, the impact of different levels of Arg and Met on the profile of myofibrillar proteins has not been studied so far, only on the indicators of body weight gain, feed conversion, carcass composition, meat quality, and immune status [1–4]. Nevertheless, Fernandes et al. [1] showed that supplementation with arginine in broilers in the initial fattening phase above the recommended level increases the weight of breast fillets and the diameter of skeletal myofibers.

In addition, those authors did not observe the effect of Arg in a diet on the protein:DNA ratio, which demonstrates that Arg does not affect the mitotic activity of the satellite cells. On the other hand, Castro et al. [3] showed that dietary supplementation with Arg leads to overall body growth with increased lean deposition. The authors of these studies argued that the increased protein content in turkey meat is likely related to increased creatine levels, an endogenous Arg metabolite involved in protein metabolism. This result partly corresponds to the results of the current authors' own research, which showed that the groups with the highest share of Arg were characterized by a slightly different profile of myofibrillar proteins, regardless of the level of Met in the diet. In another study, Zhai

et al. [2] showed that supplementing the diet of chickens with methionine (Met) resulted in an increase in muscle protein deposition, which, however, was more the result of an increase in the amount of sarcoplasmic proteins than myofibrils. Additionally, Zhai et al. [2] stated that myofibrillar hypertrophy is associated with a quantity increase of actin and myosin, which increases the strength of muscle contraction and is associated with physical activity and exercise. This myofibrillar hypertrophy could worsen the tenderness of meat [2]. This statement is in agreement with our results in relation to actin. The level of actin increased with a higher level of Arg (100% and 110%) independently of the level of Met. The opposite effect was found for myosin HC, LC1, and LC2. Bowker and Zhuang [25] found a lower level of myosin HC in broiler breast meat, with severe degrees of white striping in relation to normal meat. The results presented in Tables 6 and 7 show the impact of the studied nutritional factors, i.e., Arg and Met level, on the amount of myosin HC. These results show that the increase in the share of Arg and Met in the diet reduces the share of myosin HC in the myofibrillar protein profile in turkey muscle. Huffman et al. [11], based on a study on muscle myosin HC growth in turkeys, put forward the hypothesis that nutritional factors such as feed restriction could alter the molecular mechanisms controlling muscle growth in poultry. However, other results obtained by Li et al. [36] did not show any differences in the level of myosin and actin between the muscles of chickens with PSE defects and normal ones. Only the effect of Met as the main factor was found for the level of myosin LC2. A study by Cai et al. [28] showed that myosin LC2 was overabundant in the muscle of woody broiler breast meat. A higher level of myosin LC2 may lead to a greater concentration of fast glycolytic muscle fibers and decreased pH. This statement is partly in agreement with the results presented in Table 5 for the parameter of b* color because Rammouz et al. [23] showed a significant relationship between muscle glycolytic potential and b* value. However, these observations are not confirmed by the results of the current study for ultimate pH. Another study by Soglia et al. [37] showed that the muscle of chickens with wooden breast defects as well as white striping exhibited a lower abundance of myosin LC1 in relation to normal samples. Additionally, Mudalal et al. [38] showed that the absolute concentrations of myofilament proteins such as actin LC1 and LC3 myosin were decreased in chicken breasts with white striping defects and may indicate the degeneration process of myofilament proteins. The results presented in Table 5 showed a lower abundance of myosin LC1 as an effect of the increasing Arg level, independent of the Met level. In conclusion, the current results showed that different levels of Arg and Met in the turkey diet influenced the quantitative distribution of myofibrillar proteins. The obtained differences in the levels of different myofibrillar proteins may be the results not only of their synthesis but also of degradation during post-mortem proteolysis [39]. It has been shown that the calpain proteolytic system is responsible for this post-mortem proteolysis, and other proteases are also involved [39,40]. The activity of these enzymes is related to many different factors, i.e., $Ca^{2+}$ concentrations, pH of muscle, temperature, stress before slaughter, age of the animal, nutrition, genetics, and many other factors [41]. Research studies have demonstrated that, within duck muscles, the calpain system significantly influences the degradation of desmin and troponin-T [41]. However, the outcomes of our results indicate no discernible disparities in troponin-T degradation among the analyzed groups. This suggests that the effects of arginine (Arg) and methionine (Met) need further exploration. In the case of desmin, the results showed a significant effect of Arg and Met, and an increase in the level of Arg in the diet was associated with an increase in the amount of desmin, while an increase in Met caused its decrease. However, it should be recalled that desmin performs a variety of functions in the cell, such as ensuring the correct position of cellular organelles by creating a "lattice skeleton" connecting to the Z line, maintaining the shape and tension of the cell walls and intracellular elements, maintaining proper communication between the cell and the matrix extracellular, ensuring mechanical integration during muscle contraction and relaxation, supporting the work of tubulin and actin, sending information within cell elements, and regulating intracellular signaling and gene expression [42,43]. It seems that, in this case, the increase in Arg in the

diet had a positive effect on the increase in desmin levels and was related to its demand in the cell due to its multiple functions. However, based on the results concerning proteins with lower molecular weights (33 kDa, 32 kDa, 28 kDa, 26 kDa, 14 kDa) and the share of tropomyosin, it can be concluded that the increase in the share of Arg in the diet of turkeys had a negative impact on the processes of muscle protein degradation and, thus, post-mortem tenderization since their amounts decreased with increasing Arg in the diet. These proteins are often considered indicators of post-slaughter meat tenderization [40]. Thus, referring to the previous considerations and these results, it can be concluded that the increase in the share of Arg and Met in the diet of turkeys increases the content of some myofibrillar proteins (actinin, desmin, actin) and reduces degradation during the post-slaughter proteolysis of proteins that are considered tenderization indicators. This may affect the functional properties during processing and the sensory quality of turkey meat. The cluster analysis findings revealed that, when Arg is present at a high concentration of 110%, variations in Met levels at 35% or 45% appear to have diminished significance. However, at the Arg levels of 100% or 90%, there were no significant differences regardless of the Met level (30% or 45%).

### 5. Conclusions

The results showed that an increase in the proportion of Arg in relation to different Met levels could affect the yellowness of meat.

The obtained results indicate that the levels of Arg and Met in the diet significantly affect the profile of myofibrillar proteins in muscles, which may be the result of differential proteolysis as well as a different protein profile before slaughter. This may affect the functional properties during processing and the sensory quality of turkey meat. Multivariate statistical analyses, i.e., principal component analysis and cluster analysis, showed that individual groups have their own unique and characteristic protein profile.

The results of the cluster analysis of the myofibrillar protein profile showed that, with a high level of Arg in the diet (i.e., 110%), the level of Met 35% or 45% in the diet is unimportant.

**Author Contributions:** Conceptualization, P.K., J.J. and W.P.; methodology, P.K., J.J., W.P. and E.Ż.; software, W.P.; validation, P.K., D.J. and E.Ż.; formal analysis, P.K. and W.P.; investigation, P.K., E.Ż., P.S., D.S., A.D. and K.K.; resources, P.K.; data curation, W.P.; writing—original draft preparation, P.K. and W.P.; writing—review and editing, P.K., W.P. and E.Ż.; visualization, W.P.; supervision, P.K. and J.J.; project administration, P.K.; funding acquisition, P.K. and J.J. All authors have read and agreed to the published version of the manuscript.

**Funding:** This work was supported by the National Science Centre in Poland, Grant No. 2017/27/B/NZ9/01007. The project was financially supported (cost of proofreading and open access charge) by the Minister of Education and Science under the program entitled "Regional initiative of excellence" for the years 2019–2023, Project no. 010/rid/2018/19, amount of funding: 12.000.000 PLN.

**Institutional Review Board Statement:** The study protocol and all procedures, including the number of birds used in this study, were evaluated and approved by the Local Ethics Committee at the University of Warmia and Mazury, Olsztyn, Poland (Resolution no. 82/2017), and the birds were cared for under guidelines comparable to those laid down by EU Directive 2010/63/EU and met the guidelines approved by the institutional animal care and use committee (IACUC).

**Data Availability Statement:** The data presented in this study are available upon request from the corresponding author.

**Conflicts of Interest:** The authors declare no conflicts of interest.

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
