# Peer review of "Myofibrillar Protein Profile of the Breast Muscle in Turkeys as a Response to the Variable Ratio of Limiting Amino Acids in Feed"

_agriculture, doi:10.3390/agriculture14020197_

Round 1
Reviewer 1 Report
Comments and Suggestions for Authors
The results of this experiment indicate that the levels of Arg and Met in the diet significantly affect the profile of myofibrillar proteins in muscles. However, it is necessary to provide the chemical composition and ingredients for the experimental diets, particularly with regard to the amino acid profile. While the experimental design is a 3x2 factorial, the interaction between the two factors is not reported in the table. There was no mention of the interaction between the two factors in the results section. Every table should show p-values. Discussion must be improved to refer to the main effect of Arg and Met on myofibrillar protein profile. The conclusion section should be succinct in order to encompass all of the experimental findings.
Comments on the Quality of English Language
English language fine
Author Response
Response to Reviewer #1
R/remark
A/answer
R/The results of this experiment indicate that the levels of Arg and Met in the diet significantly affect the profile of myofibrillar proteins in muscles.
However, it is necessary to provide the chemical composition and ingredients for the experimental diets, particularly with regard to the amino acid profile.
A/We included now two tables (please see Table 1 and Table 2) showing analyzed content of Lys, Met and Arg in the experimental diets as well ingredients composition of basal diets.
R/While the experimental design is a 3x2 factorial, the interaction between the two factors is not reported in the table.
A/We would like to point out that the interaction effect of both main factors is presented in Table 4 and Table 6. This is highlighted in the table titles in brackets.
R/There was no mention of the interaction between the two factors in the results section.
A/The interaction effect for meat quality characteristics in the Results section is described in lines 183-186. It was significant only for the color parameter b*. However, with respect to myofibrillar proteins, the interaction effect presented in Table 6 is commented in the text on lines 222-231.
R/Every table should show p-values.
A/In tables p-values are added.
R/Discussion must be improved to refer to the main effect of Arg and Met on myofibrillar protein profile.
A/Thank you for your remark. We refined the description of the main effect for Arg and Met. It is described in lines 195-199 and 204-206 respectively.
R/The conclusion section should be succinct in order to encompass all of the experimental findings.
A/Thank you for your comment and, in accordance with your suggestion, we have improved the Conclusion subsection, referring to the most important research results.
Reviewer 2 Report
Comments and Suggestions for Authors
1. Most of commercial turkeys are slaughter after 9 weeks of age. The authors need to justify why the study set up at 42 days for breast muscle sample collection?
2. In line 87-88, “eight turkeys representing the group's average body weight from each treatment were sacrificed by cervical dislocation, and the right breast muscle (pectoralis major) was collected, if so it means not randomly sampling and how can the authors conclude the effects by treatments?
3. In SDS page for protein profile, I supposed the methodology is based on the same protein loading. However, in real muscle tissues, the ash and water may influence total protein fraction. The authors may need to show a crude component composition ate least including water, ash, protein, and lipid.
4. In statistics, the study was designed as two-way ANOVA (3 x 2 factorial design). However, all tables did not show the complete statistic including main effects and interaction. This is probably due to too many measurement items. At least, in the footnote of each table, the interaction P-value should be shown to make the following LSD comparison rationally.
5. The breast muscle is a big tissue and only a meat sample of 150 mg was used for protein extraction. It may cause variable by different parts of the muscle. The authors need to specify the part of muscle.
Comments on the Quality of English Language
Minor editing of English language required
Author Response
Response to Reviewer #2
R/remark
A/answer
R/Most of commercial turkeys are slaughter after 9 weeks of age. The authors need to justify why the study set up at 42 days for breast muscle sample collection?
A/Thank you for this comment. Indeed, commercial turkeys are usually slaughtered at later age however one of the important goals of this research was to investigate effect of different proportion AA at the early stage of the bird’s growth as this may contribute to further development of indices determining overall quality of breast muscles. In other words, essential AA, play a key role in the regulation of homeostasis in the whole biological system, including body maintenance, growth, immunity and reproduction and the initial phase of growth of birds may be crucial. A number of evidences was already found in this regard by other Authors and in our recent studies, in which we evidenced that the diets of increased Arg, Lys, and Met fed to turkeys over the first 4-wk period of growth and exposed to different challenge factors (including C. perfringens challenge) increased significantly the transcripts levels not only of tight junction proteins genes but also selected genes encoding nutrient transporters (Increased arginine, lysine, and methionine levels can improve the performance, gut integrity and immune status of turkeys but the effect is interactive and depends on challenge conditions Vet. Res., 53 (2022)). Corresponding to the above we raised this point in the manuscript, please see lines 61-62.
R/In line 87-88, “eight turkeys representing the group's average body weight from each treatment were sacrificed by cervical dislocation, and the right breast muscle (pectoralis major) was collected, if so, it means not randomly sampling and how can the authors conclude the effects by treatments?
A/Thank you for this remark. Just to clarify, this is standard procedure in our Laboratory or elsewhere, by selection birds for sampling it is important to avoid taking those outstanding (in each separate pen) to a high extent, which may suffer from disorders. Therefore, they are all weighted before sampling, and we considered a group average. In particularly, that turkeys at 42 days of age may still suffer from disorders associated with yolk sac absorption or so, which could contribute to number of physiological indices.
R/In SDS page for protein profile, I supposed the methodology is based on the same protein loading. However, in real muscle tissues, the ash and water may influence total protein fraction. The authors may need to show a crude component composition ate least including water, ash, protein, and lipid.
A/Thank you for this comment and we agree that the chemical composition of muscle tissue and protein content is significantly influenced by the water, ash and fat content. However, we would like to emphasize that the study analysed the profile of proteins building myofibrils in turkey muscles. Since the share of myofibrillar proteins in each group is presented in the form of percentages of individual fractions within the group, the total protein content is not important here. Furthermore, the amount of protein loaded onto the electrophoretic tracks in each sample was identical because it was corrected based on the protein content of the muscle tissue. This results from methodological assumptions. This is described in subsection 2.5. Extraction of muscle protein line 134.
R/In statistics, the study was designed as two-way ANOVA (3 x 2 factorial design). However, all tables did not show the complete statistic including main effects and interaction. This is probably due to too many measurement items. At least, in the footnote of each table, the interaction P-value should be shown to make the following LSD comparison rationally.
A/Thank you for this attention and we would like to mention that the main effects are presented in Table 3 and Table 5. The interaction effect of the main factors is presented in Table 4 and Table 6. According to your suggestion, the p-value was added.
R/The breast muscle is a big tissue and only a meat sample of 150 mg was used for protein extraction. It may cause variable by different parts of the muscle. The authors need to specify the part of muscle.
A/Samples were collected in an identical manner from each individual mid-muscle sample. Following your comment, we have added it to the text (line 134).
Reviewer 3 Report
Comments and Suggestions for Authors
Reviewer’s comments
· Ln 12: Kindly rephrase. The Arginine/Lysine ratio was not investigated in this study
· Ln 69: Provide the average body weight of the experimental birds
· Ln 70: Kindly provide the exact conditions implemented for “temperature, humidity, and lighting programs”
· Ln 74: Provide the calculated and analyzed values for the experimental diet’s composition
· Ln 75: How was the diet balanced to achieve an isocaloric diet? Also, were the diets formulated to maintain isonitrogeneity across diets, especially given the role of arginine in nitrogen metabolism?
· Table 1: Given that the study is a factorial arrangement, kindly provide the results showing the interaction effects
· Table 2: Replace the comma with a decimal point in all the values presented.
· Table 2: There is no mention of Liz ratio. Correct this: “Nutritional effect of Arg-Met/Liz ratio”
· Table 3: It should be revised to indicate the significant main effects of both dietary treatments investigated or their interaction.
· Ln 169-170: No significant main effects of Met is given in Table 3 as reported in the sentence below:
o “In turn, the influence of the level of Met in the diet was found for the levels of Myosin HC, Desmin, Tropomyosin, TnC and Myosin LC2 (Table 3).”
· Kindly discuss the significance of the finding that “increase in the proportion of Arg in relation to different Met levels could affect the yellowness of meat.” What does Arg-Met-induced yellowing translate to meat quality?
· Limited information (pH and color) has been provided on the meat quality attributes investigated. Provide additional information for production (growth rate, carcass traits) and meat quality attributes (proximate composition, tenderness, water-holding capacity, muscle myopathy, etc) that will support the findings of the present study
· Ln 223: Change “lover” to “lower”
· Is there any notable influence of the dietary treatments (Arg/Met) on the expression of the myofibrillar protein based on the molecular weight distribution of individual protein bands?
· 273-276: Provide clear reasons to explain the differential effects of Arg and Met on the muscle myofibrillar protein profile
· Ln 335: From this study, what recommendations should be considered given that “increase in the share of Arg in the diet of turkeys had a negative impact on the processes of muscle protein degradation, and thus post-mortem tenderization”?
· Kindly rewrite the conclusion section to communicate the important findings of the study, their significance, and possible areas for further study. Avoid repeating the contents of the result section.
Comments on the Quality of English Language
The quality of written English is sufficient for proper interpretation and comprehension by readers. Additional attention to misspelt words and characters would be beneficial.
Author Response
Response to Reviewer #3
R/remark
A/answer
R/Ln 12: Kindly rephrase. The Arginine/Lysine ratio was not investigated in this study
A/It is rewritten now, please see lines 12-13.
R/Ln 69: Provide the average body weight of the experimental birds
A/It is provided now. Please see lines 166-171, as well as their discussion (lines 262-265).
R/Ln 70: Kindly provide the exact conditions implemented for “temperature, humidity, and lighting programs”
A/The detailed programs were provided (lines 71-77).
R/Ln 74: Provide the calculated and analyzed values for the experimental diet’s composition
A/It is now provided in tables 1 and 2.
R/Ln 75: How was the diet balanced to achieve an isocaloric diet? Also, were the diets formulated to maintain isonitrogeneity across diets, especially given the role of arginine in nitrogen metabolism?
A/Thank you for this remark, the diets were formulated to be isocaloric and isonitrogenity using formulation programme (calculation of nutrients content), and then analytically confirmed regarding AA, protein, energy confirmed. The details can be found now in tables 1 and 2.
R/Table 1: Given that the study is a factorial arrangement, kindly provide the results showing the interaction effects
A/We would like to underline that the interaction effect of both main factors is presented in Table 4 and Table 6. The interaction effect for meat quality characteristics in the Results section is described in lines 182-185. It was significant only for the color parameter b*. However, with respect to myofibrillar proteins, the interaction effect presented in Table 6 is commented in the text on lines 221-230.
R/Table 2: Replace the comma with a decimal point in all the values presented.
A/As noted, commas in Table 2 (now Table 4) have been replaced with dots.
R/Table 2: There is no mention of Liz ratio. Correct this: “Nutritional effect of Arg-Met/Liz ratio”
A/It is corrected.
R/Table 3: It should be revised to indicate the significant main effects of both dietary treatments investigated or their interaction.
A/It is corrected. Thank you for your suggestion.
R/Ln 169-170: No significant main effects of Met is given in Table 3 as reported in the sentence below:
o “In turn, the influence of the level of Met in the diet was found for the levels of Myosin HC, Desmin, Tropomyosin, TnC and Myosin LC2 (Table 3).”
A/Thank you very much for your remark. The missing markings have been completed and were probably removed when formatting the table when transferring the publication to the template.
R/Kindly discuss the significance of the finding that “increase in the proportion of Arg in relation to different Met levels could affect the yellowness of meat.” What does Arg-Met-induced yellowing translate to meat quality?
A/The color parameter b* is significantly related to the quality of meat and its defects. The aspect is highlighted in the discussion in lines 279-89. This hypothesis was then supported by literature data. In our opinion, this issue has been sufficiently described.
R/Limited information (pH and color) has been provided on the meat quality attributes investigated. Provide additional information for production (growth rate, carcass traits) and meat quality attributes (proximate composition, tenderness, water-holding capacity, muscle myopathy, etc) that will support the findings of the present study
A/Thank you for this remark. Initially, the issue of pH was not discussed extensively because no significant statistical effect of the tested factors on this feature was demonstrated. However, taking into account your recommendation, we have expanded the discussion and it is presented in lines 267-276.
R/Ln 223: Change “lover” to “lower”
A/It is improved.
R/Is there any notable influence of the dietary treatments (Arg/Met) on the expression of the myofibrillar protein based on the molecular weight distribution of individual protein bands?
A/This issue is difficult to explain. Because determining expression would rather refer to the study of gene expression in living animals. However, this has not been investigated. Only the profile of myofibrillar proteins in slaughtered meat was examined.
R/273-276: Provide clear reasons to explain the differential effects of Arg and Met on the muscle myofibrillar protein profile
A/As shown in the discussion on lines 260-288, there are still no clear study results. However, it was indicated that some results suggest that supplementing the poultry diet with higher levels of Arg may result in higher muscle tissue growth.
R/Ln 335: From this study, what recommendations should be considered given that “increase in the share of Arg in the diet of turkeys had a negative impact on the processes of muscle protein degradation, and thus post-mortem tenderization”?
A/It should be emphasized that there are no similar studies. Our own research indicates a significant impact of Arg and Met in the turkey diet on the muscle protein profile. However, it is too early to formulate practical recommendations.
R/Kindly rewrite the conclusion section to communicate the important findings of the study, their significance, and possible areas for further study. Avoid repeating the contents of the result section.
A/Thank you for your suggestion. We have improved this section.
Round 2
Reviewer 1 Report
Comments and Suggestions for Authors
All comment was incorporated into the manuscript file.

Author Response
Dear Reviewer
Thank you for your comments on our publication. All remarks allowed us to improve the quality of the prepared study. All minor comments have been corrected and marked in red in the manuscript text.
According to your comment, we have prepared the new Conclusion, which is more compact in our opinion.
We have corrected the headings of tables 3 and 4 and 5 and 6 (now tables 4, 5, 6 and 7). The previous heading information were not very precise and could make the data difficult to read.
Please accept the tables layout as in the manuscript as-is. By combining tables 3 and 4 or 5 and 6 (now 4 and 5 and 6 and 7) would make them too large and make their analysis difficult for the reader.
Thank you for understanding.
Reviewer 3 Report
Comments and Suggestions for Authors
Rephrase Line 22-24: Rephrase/rewrite for clarity and provide information of the specific proteins increased/decreased in the study.
Table 1 & 2: Kindly provide which is the calculated or analyzed values on the tables provided. In Table 1, Row 4; check Arg90Met45 and correct to the right treatment
“The increase in the dietary ratio of Arg and Met it increases the breast meat content of some 22 myofibrillar proteins (actinin, desmin, actin) and reduces degradation during post-slaughter proteolysis of proteins considered as tenderization indicators.”
Ln 15: correct “detrmined” to “determined”
Ln 166: No results have been provided for the production indices of this study (body weight, feed intake, FCR) for a nutrition-based study. Kindly provide the necessary data. A discussion on body weight attributes of turkeys at 42 days without showing any dataset does not suffice.
On the table sub-headings, change “P” to “P value"
Table 5: Provide the name for Trait #12
Table 6: correct “Different Arg and Met” to “Different Arg and Met 35”
In the discussion section, avoid inserting the tables/figures captions; and repeated interpretation of your results, which was already covered in the “result section”. You should focus on the meaning and implications of the data that you have already reported in your results section.
Comments on the Quality of English Language
The quality of English language in the paper is okay. Kindly pay attention to incorrect word spellings throughout the paper
Author Response
Authors are grateful for further remarks to improve our paper, we addressed all raised points. All changes are marked in red.
R/Rephrase Line 22-24: Rephrase/rewrite for clarity and provide information of the specific proteins increased/decreased in the study.
A/The indicated lines are rewritten
R/Table 1 & 2: Kindly provide which is the calculated or analyzed values on the tables provided. In Table 1, Row 4; check Arg90Met45 and correct to the right treatment
A/The Tables are improved.
R/“The increase in the dietary ratio of Arg and Met it increases the breast meat content of some 22 myofibrillar proteins (actinin, desmin, actin) and reduces degradation during post-slaughter proteolysis of proteins considered as tenderization indicators.”
A/This sentence is changed.
R/Ln 15: correct “detrmined” to “determined”
A/It is corrected.
R/Ln 166: No results have been provided for the production indices of this study (body weight, feed intake, FCR) for a nutrition-based study. Kindly provide the necessary data. A discussion on body weight attributes of turkeys at 42 days without showing any dataset does not suffice.
A/We presented performance response of birds as a result of treatments, please see a new Table3.
R/On the table sub-headings, change “P” to “P value
A/It is changed.
R/Table 5: Provide the name for Trait #12
A/The missing expressions is added (please see Table 6 now).
R/Table 6: correct “Different Arg and Met” to “Different Arg and Met 35”
A/We have corrected it (please see Table 7 now).
R/In the discussion section, avoid inserting the tables/figures captions; and repeated interpretation of your results, which was already covered in the “result section”. You should focus on the meaning and implications of the data that you have already reported in your results section.
A/Thank you very much for this remark. We removed the repeated interpretation of our results from the discussion section.